# Evaluation of the clinical performance of a magnetic force-assisted electrochemical immunoassay for the detection of SARS-CoV-2 antigens

**Sung Jin Jo**[1], **Sang-hyun Shin**[1], **Jungrok Kim**[1], **Seungok Lee**[2], **Jehoon Lee**[1]*

**1** Department of Laboratory Medicine, Eunpyeong St. Mary's Hospital, College of Medicine, The Catholic University of Korea, Seoul, Republic of Korea, **2** Department of Laboratory Medicine, Incheon St. Mary's Hospital, College of Medicine, The Catholic University of Korea, Incheon, Republic of Korea

* lyejh@catholic.ac.kr

**Data Availability Statement:** All relevant data are within the manuscript.

**Funding:** This work supported by BBB. The funders had no role in study design, data collection and

## Abstract

Rapid antigen (Ag) tests for the detection of severe acute respiratory syndrome coronavirus 2 (SARS-CoV-2) provide quick results, do not require specialized technical skills or infrastructure, and can be used as a point-of-care method to prevent the spread of coronavirus disease (COVID-19). The performance of a magnetic force-assisted electrochemical immunoassay-based test, namely the MARK-B COVID-19 Ag test (BBB, Sungnam, Republic of Korea), was evaluated using 170 nasopharyngeal swab specimens and compared to that of RT-PCR and commercial rapid Ag test (STANDARD Q COVID-19 Ag Test, SD Biosensor, Suwon-si, Republic of Korea). The overall sensitivity and specificity of the MARK-B test were 90.0% (95% CI 79.4%–96.2%) and 99.0% (95% CI 95.0%–99.9%), respectively, with a kappa coefficient of 0.908. The correlations between the electrical current values of MARK-B and the Ct values of RT-PCR were −0.898 (*E* gene, 95% CI −0.938 to −0.834) and −0.914 (*RdRp* gene, 95% CI −0.948 to −0.860), respectively. The limit of detection of the MARK-B was measured using the viral culture reference samples and found to be $1 \times 10^2$ pfu/mL. The magnetic force-assisted electrochemical immunoassay-based Ag test can be used to rapidly detect SARS-CoV-2 infections, and the corresponding fully automated portable device can provide easy readability and semi-quantitative results.

## Introduction

Reverse-transcription polymerase chain reaction (RT-PCR) is considered the standard method for the diagnosis of coronavirus disease (COVID-19) because of its high sensitivity and specificity [1–3]. However, RT-PCR has the following disadvantages: it is expensive in terms of the cost of testing, requires established laboratory facilities, and involves a relatively longer test time [4]. Furthermore, COVID-19 testing in asymptomatic populations is increasing, which has led to an increased gap between the test demand and laboratory capacity [5]. Notably, rapid antigen tests for severe acute respiratory syndrome coronavirus 2 (SARS-CoV-2) can be

analysis, decision to publish, or preparation of the manuscript.

**Competing interests:** The authors have declared that no competing interests exist.

processed at the point of care, and the results are available within minutes. Moreover, rapid antigen testing is not dependent on advanced technical skills or infrastructure [6, 7]. Despite these advantages, rapid antigen tests are not recommended for the clinical diagnosis of COVID-19 owing to their relatively low sensitivity [8]. Recently evaluated rapid antigen tests for SARS-CoV-2 displayed a wide range of sensitivity (i.e., 17.5%–98%) with compatible specificity (i.e., 98%–100%) [6, 7, 9–17]. Nevertheless, if sufficient accuracy is obtained, then rapid antigen tests may replace RT-PCR based tests as a means for triaging or may play a role in rapid determination. Furthermore, rapid Ag tests can be easily delivered to the site, and they can be used for the mass screening of COVID-19 patients and in a more targeted manner at airports, schools, and international/regional borders [18, 19].

The MARK-B COVID-19 Ag test (MARK-B, BBB Inc., Sungnam, Republic of Korea) is a portable point-of-care device based on magnetic force-assisted electrochemical immunoassay (MESIA) designed to detect SARS-CoV-2 nucleocapsid antigens. Notably, this newly developed immunoassay, MESIA, is reportedly suitable for measuring low concentrations of proteins for cancer detection and has the potential to detect other protein biomarkers [20, 21]. MESIA may provide enhanced sensitivity for point-of-care devices that involve the use of immunoassays to detect SARS-CoV-2. Herein, we evaluated the clinical performance of the MESIA-based MARK-B test for the first time, compared to that of RT-PCR and a commercially available rapid antigen (Ag) test. Rapid Ag testing with improved accuracy may serve a variety of purposes in preventing the spread of COVID-19.

## Methods

### Clinical sample collection

Nasopharyngeal swab (NP) specimens were collected from patients with COVID-19 symptoms. These patients were subjected to an RT-PCR test for SARS-CoV-2 by clinicians in Eunpyeong St. Mary's Hospital or Incheon St. Mary's Hospital, Republic of Korea, between October 2020 and February 2021. NP specimens were placed in a universal transport medium (UTM; T-SWAB TRANSPORT CTM, Noble Biosciences, Hwaseong-si, Republic of Korea), and they were initially evaluated using RT-PCR. The remnant NP specimens were stored at −70˚C, and they were used to evaluate the clinical performance of the MARK-B tests. Clinical information, such as symptoms and days from onset of symptoms (DFOS), was retrieved from medical records. This study was approved by the Institutional Review Board of the Catholic University of Korea, Republic of Korea (XC21DDDT0025), and the informed consent was waived.

### MESIA for SARS-CoV-2 antigen detection

The MARK-B test is an *in vitro* medical device that is based on principles of MESIA and intended for the qualitative and semi-quantitative detection of the SARS-CoV-2 nucleocapsid antigens. When a sample is loaded into a cartridge, it flows into the microchannel, containing gold-coated magnetic nanoparticles and an electrochemical sensor. The analytes form immune complexes under external magnetic fields. The targeted antigen level is measured based on the electrochemical redox signal of gold-coated magnetic nanoparticles conjugated with the antibodies. The electrical signal is measured using a fully automated portable device, the MARK-B 1 Analyzer (BBB), and the result is determined in 10 min based on the cut-off value that has been set as per the manufacturer's instructions. The preset cut-off value is determined for each lot once manufactured based on the mean and the standard deviation of the electrical signal obtained from negative samples. The QR code on each cartridge contains the information and the analyzer can recognize the cut-off of each cartridge when the QR code is scanned. The

assay provides a result that indicates the absence or presence of SARS-CoV-2, along with the quantity of the captured targets from the specimen, measured using the electrical signals.

The MARK-B tests using NP specimens were performed according to the manufacturer's instructions, which recommended the direct use of a nasopharyngeal swab or an aliquot of UTM. The previously stored UTM samples were thawed at room temperature for 30 min and vortexed for 10 s. Manufacturer recommends using 400 μL of UTM to extract SARS-CoV-2 nucleocapsid proteins from the specimen by mixing it with the extraction buffer provided with the product. However, the remnant NP specimen after RT-PCR was not enough. Therefore, the volume of the extraction buffer added was 200 μL for MARK-B™ COVID-19 Ag, and the sample was 1:1 diluted after UTM is added. Three drops of the specimen were applied to the cartridge, and the device was subjected to operation. Test results above the lot-specific cut-off value (i.e., 8.325 μA, Lot No. 0AAD0060B120131) were considered positive.

## RNA extraction and real-time RT-PCR for SARS-CoV-2 detection

Viral RNA extracted from the nasopharyngeal specimens (200 μL each) was performed using the NX-48 viral NA kit (Genolution, Seoul, Republic of Korea) and the Nextractor NX-48 system (Genolution). Nucleic acids were extracted according to the manufacturer's instructions. SARS-CoV-2 was amplified using real-time RT-PCR, with a commercial Real-Q 2019-nCoV Detection Kit (BioSewoom, Seoul, Republic of Korea). The *E* and *RdRp* genes of SARS-CoV-2 were amplified over 40 cycles using the Applied Biosystems 7500 RT-PCR system (Thermo Fisher Scientific, Waltham, MA). Samples were considered as SARS-CoV-2-positive when both targets of viral RNA had been amplified under a cycle threshold (Ct) of 38.0.

## Rapid antigen test for SARS-CoV-2

The STANDARD Q COVID-19 Ag Test (SD Biosensor, Suwon-si, Republic of Korea) was used for the detection of SARS-CoV-2 antigens in NP specimens. Notably, the STANDARD Q COVID-19 Ag (SDQ) and the MARK-B tests were conducted concurrently. The stored UTM samples were thawed at room temperature for 30 min and then vortexed for 10 s. The manufacturer recommends using 350 μL of UTM and 350 μL of extraction buffer for the dilution factor 1:1. As described earlier, the remnant NP specimens was not enough, 200 μL of the NP specimen from stored UTM was mixed with the extraction buffer provided in the test kit (1:1). Three drops of the extracted specimen were applied to the test device, and the result was recorded within 15–30 min. The test result was considered positive when both the control and the test lines were colored.

## Limit of detection with viral culture samples

To compare the limit of detection (LOD) between two rapid Ag kits, a serial dilution of SARS-CoV-2 samples were used. Vero E6 cells (Korean Cell Line Bank, Seoul, Korea) were cultured and incubated with the SARS-CoV-2 strain (BetaCoV/Korea/KCDC03/2020: NCCP 43326, National Culture Collection for Pathogens, Osong, Korea). The viral concentration was quantified to be equivalent to $6.5 \times 10^5$ pfu/ml and samples were serially diluted to $1.0 \times 10^5$ pfu/ml, $1 \times 10^4$ pfu/ml, $1 \times 10^3$ pfu/ml, $4 \times 10^2$ pfu/ml, $2 \times 10^2$ pfu/ml, and $1 \times 10^2$ pfu/ml. MARK-B and SDQ tests were repeated five times for each diluted sample according to the manufacturer's instructions. Cell culture procedures were performed according to biosafety level 3 (BSL-3) conditions.

**Table 1. Sensitivity and specificity of MARK-B and SDQ compared to those of real-time RT-PCR.**

| | | Real-time RT-PCR | | |
|---|---|---|---|---|
| | | **Positive** | **Negative** | |
| MARK-B | Positive | 54 | 1 | MARK-B sensitivity = 90.0% (95% CI = 79.4%–96.2%) |
| | Negative | 6 | 109 | MARK-B specificity = 99.0% (95% CI = 95.0%–99.9%) |
| | | | | *k* value = 0.908 |
| SDQ | Positive | 34 | 0 | SDQ sensitivity = 56.7% (95% CI = 43.2%–69.4%) |
| | Negative | 26 | 110 | SDQ specificity = 100% (95% CI = 96.7%–100%) |
| | | | | *k* value = 0.628 |

CI, confidence interval.

## Statistical analysis

Statistical analysis was performed using the MedCalc software ver 19.6.1 (MedCalc Software, Ostend, Belgium). Figures were created using GraphPad Prism ver 9.1.2 (GraphPad Software, San Diego, CA). The Kolmogorov–Smirnov test was used to assess the normality of the distribution. The correlation between the Ct values of the *E* and *RdRp* genes obtained via RT-PCR and the measured MARK-B electrical current values was evaluated using Pearson's correlation coefficient (r). Receiver operating characteristic (ROC) curve analysis was performed to evaluate the MARK-B lot-specific cut-off value with clinical samples. Cohen's kappa coefficient (*k*) was used to assess the inter-rater reliability among SARS-CoV-2 antigen assays. Notably, $k < 0$ indicated no agreement, while agreement was considered to be slight for $k = 0$–0.20, fair for $k = 0.21$–0.40, moderate for $k = 0.41$–0.60, and substantial for $k = 0.61$–0.80; $k = 0.81$–1 represented an almost perfect agreement. The statistical significance threshold was set at $P < 0.05$.

## Results

In total, 60 samples were analyzed using RT-PCR and were considered SARS-CoV-2-positive, while 110 samples were considered SARS-CoV-2-negative. Among the 60 samples that were confirmed as positive, 24 NP specimens had been submitted 0–3 DFOS, 24 NP specimens had been collected 4–7 DFOS, and 12 had been obtained >8 DFOS.

The comparison of the MARK-B and SDQ test results with the RT-PCR results is summarized in Table 1. Overall, MARK-B was characterized by 90.0% sensitivity (95% CI, 79.4%–96.2%) and 99.0% specificity (95% CI, 95.0%–99.9%) for SARS-CoV-2 Ag detection, and there were six false negatives. The *k* value of MARK-B compared to RT-PCR was 0.908. Based on the *RdRp* Ct value ranges (Table 2), MARK-B displayed 100% (i.e., 53/53) sensitivity for

**Table 2. Sensitivity of MARK-B and SDQ according to Ct range and target genes of real-time RT-PCR.**

| | | *E* gene | | | *RdRp* gene | | |
|---|---|---|---|---|---|---|---|
| | | **< 25** | **25–30** | **> 30** | **< 25** | **25–30** | **> 30** |
| MARK-B | Positive | 42 | 11 | 1 | 40 | 13 | 1 |
| | Negative | 0 | 2 | 4 | 0 | 0 | 6 |
| | Sensitivity (95% CI) | 100% (91.5%–100%) | 84.6% (54.5%–98.0%) | 20.0% (0.5%–71.6%) | 100% (91.1%–100%) | 100% (75.2%–100%) | 14.2% (0.3%–57.8%) |
| SDQ | Positive | 32 | 2 | 0 | 33 | 1 | 0 |
| | Negative | 10 | 11 | 5 | 7 | 12 | 7 |
| | Sensitivity (95% CI) | 76.1% (60.5%–87.9%) | 15.3% (1.9%–45.4%) | 0% (0%–52.1%) | 82.5% (67.2%–92.6%) | 7.6% (0.1%–36.0%) | 0% (0%–40.9%) |

CI, confidence interval.

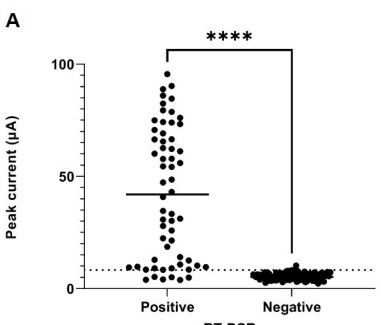 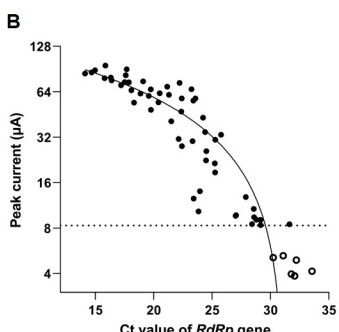 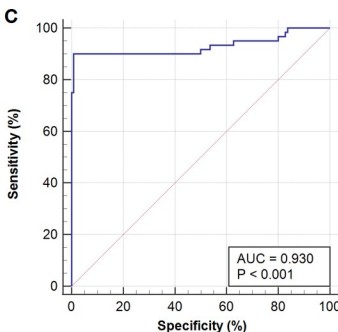

**Fig 1. RT-PCR and MARK-B test results.** (A) Peak current values of RT-PCR-positive and RT-PCR-negative samples. (B) Correlation between peak current values (MARK-B, log2 scale) and the Ct values of the *RdRp* gene (60 RT-PCR-positive samples; ○, MARK-B false negative). (C) Receiver operating characteristic (ROC) curve analysis. The MARK-B test indicated an area under the ROC curve value of 0.930 (95% CI 0.880–0.963). Dotted line: lot-specific cut-off (8.325 μA), ****P < 0.0001.

specimens characterized by Ct ≤ 30, but 14.2% (i.e., 1/7) sensitivity for specimens with Ct > 30. Furthermore, the sensitivity of the MARK-B test was estimated for RT-PCR-positive samples according to DFOS. MARK-B displayed a sensitivity of 91.6% (i.e., 22/24) for samples taken 0–3 DFOS, 91.6% (i.e., 22/24) for samples obtained 4–7 DFOS, and 83.3% (10/12) for specimens sampled ≥ 8 DFOS.

The mean electrical current values of MARK-B for RT-PCR-positive samples (i.e., 42.34 (SD 29.31) were significantly higher (P < 0.001, 95% CI = −42.43 to −31.4; Fig 1A) than those of RT-PCR-negative samples (i.e., 5.42 (SD 1.48)). The analysis of the correlation between the Ct values of the *E* gene and the electrical current values of MARK-B revealed a value of r = −0.898 (95% CI −0.938 to −0.834, P < 0.001), and that between the Ct values of the *RdRp* gene and the MARK-B values revealed a value of r = −0.914 (95% CI −0.948 to −0.860, P < 0.001; Fig 1B). There were six false-negative cases with MARK-B (Fig 1B), and their values were below the lot-specific cut-off value (i.e., 8.325). To evaluate the MARK-B lot-specific cut-off value with clinical samples, an ROC curve analysis was conducted. For the 170 clinical samples, the highest efficiency was estimated at a cut-off of 8.235 with an area under the curve of 0.930 (95% CI 0.880–0.963; Fig 1C). The sensitivity and specificity of the MARK-B test with a clinically estimated cut-off (8.235) yielded the same results as when the lot-specific cut-off value was applied.

Overall, the SDQ test was characterized by 56.7% sensitivity (95% CI 43.2%–69.4%) and 100% specificity (95% CI 96.7%–100%) for SARS-CoV-2 Ag detection. There were 26 false negatives, and the *k* value was 0.628 for SDQ and RT-PCR tests (Table 1). The sensitivity of the SDQ test decreased markedly for specimens with Ct ≥ 25 and showed 0% sensitivity for specimens with Ct > 30. There were 10 cases of weak positives with SDQ tests. Notably, MARK-B yielded positive results for the samples corresponding to the 10 weak positive cases. The mean Ct value of the *RdRp* gene of MARK-B-positive/SDQ-positive samples was 18.5 (SD 2.8), that of MARK-B-positive/SDQ-weak positive was 22.4 (SD 1.6), that of MARK-B-positive/SDQ-negative was 26.4 (SD 2.5), and that of MARK-B-negative/SDQ-negative was 31.8 (SD1.1) (Fig 2). There were significant differences between the mean Ct values of the *RdRp* gene in each of these four groups (Fig 2).

The LOD of the two rapid kit was tested with reference viral culture samples. Cultured viral samples were diluted to six concentrations. The measured LOD of MARK-B was 1 x 10$^2$ pfu/mL, and SDQ was 1.0 x 10$^4$ pfu/mL, respectively (Table 3).

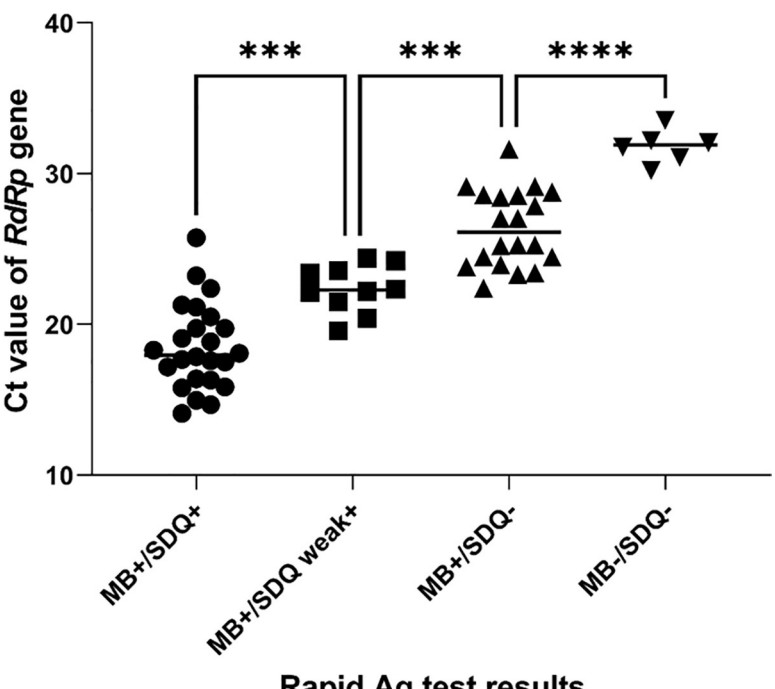

**Fig 2. MARK-B and SDQ results compared to the Ct values (*RdRp* gene) of RT-PCR-positive samples.** MB: MARK-B COVID-19 Ag test, SDQ: STANDARD Q COVID-19 Ag, +: positive, -: negative, ***P < 0.001, ****P < 0.0001.

## Discussion

In the present study, the performance of the MESIA-based Ag test MARK-B was evaluated using clinical samples and compared with the performance of RT-PCR and commercial Ag tests. The MARK-B tests displayed 90.0% sensitivity and 99.0% specificity across the samples. Moreover, the MARK-B test exhibited 100% sensitivity for samples whose RT-PCR results were Ct < 25 and 84.6% sensitivity for both *E* and *RdRp* genes (Ct = 25–30). Additionally, it was characterized by 95.9% overall agreement and a 0.908 *k* value, indicating an extremely good match between the results of RT-PCR and the MARK-B test. Both the commercial and the MARK-B Ag tests exhibited higher sensitivity for Ct values <25, and the sensitivity decreased as Ct values increased. The SDQ test reportedly exhibits a sensitivity above 95% for Ct values <25 and 53.9–62.1% for Ct value >25 [7, 22–25]. In the present study, SDQ tests displayed 76.1% sensitivity for Ct values <25 and declined sharply for Ct value >25. This study was conducted using remnant specimens; therefore, the storage status of the samples and the use of UTM might have affected the sensitivity of the rapid tests. In general, studies that involved the use of UTM reported lower sensitivity than studies that involved the use of NP

**Table 3. Comparison of the limit of detection for MARK-B and SDQ with cultured viral samples.**

| | | Virus concentration (pfu/mL) | | | | | | |
|---|---|---|---|---|---|---|---|---|
| | | Negative | 1 x 10² | 2 x 10² | 4 x 10² | 1 x 10³ | 1 x 10⁴ | 1 x 10⁵ |
| MARK-B | Mean (µA) | 6.70 | 8.782 | 9.615 | 10.727 | 18.917 | 39.896 | 61.816 |
| | SD | 0.67 | 0.24 | 0.49 | 1.03 | 2.11 | 2.41 | 5.37 |
| SDQ | | Negative | Negative | Negative | Negative | Negative | Weak positive | Positive |

specimens [7, 13, 22, 24, 25]. All RT-PCR-positive samples were maintained under freezing conditions at −70˚C before Ag tests, and few samples were refrozen after subjection to thawing once. Notably, viral accessibility can be influenced by specimen storage conditions such as the freeze-thaw cycle. One freeze-thaw cycle increased the Ct value to 0.41, and two cycles of freezing and thawing increased the Ct value to 0.82 [26]; thus, more freeze-thaw cycles tended to increase the Ct value [27].

The MARK-B tests are automatically conducted using a portable device analyzer that regulates the magnetic field and measures electrochemical signals. The qualitative results were determined based on a cut-off value, and the measured electrochemical signals were also presented. The ROC curve analysis with clinical samples demonstrated that the lot-specific cut-off value was appropriately established. The use of visual readout rapid Ag tests can produce ambiguous results at low antigen concentrations, while instrument-based Ag tests yield results with clarity. For instance, when interpreting the visual readout Ag test, the presence of a line, regardless of the faintness of the line, indicates a positive result. There were 10 weak-positive cases within the SDQ results and 20 cases of MARK-B-positive/SDQ-negative results, and there were significant differences between these sample groups. Notably, the MARK-B device provided readability for clinical samples with higher Ct values, and the MESIA technique improved the sensitivity of the immunoassay for detecting SARS-CoV-2 antigens. The LOD measurement of two rapid Ag tests were conducted to verify the sensitivity differences between the rapid Ag tests using viral culture reference samples. In line with tests using clinical samples, MARK-B tests showed a LOD at approximately $10^2$ times lower viral concentrations than SDQ.

Furthermore, the electrical current values of MARK-B and the Ct values of RT-PCR were highly correlated. We plotted log2(current) values vs. Ct values and fitted with a line. While the signal was saturated at the concentrations higher than Ct 20, the log2(current) vs. Ct showed a linear relationship in the range from Ct 20 to Ct 33, as shown Fig 1B. These results indicate that the electrochemical signals measured with the MESIA are proportional to the concentration of antigens, which suggests that MESIA can help provide reliable semi-quantitative results in conditions where RT-PCR is not available.

In conclusion, the MARK-B test, a MESIA-based rapid Ag test, showed higher sensitivity compared to commercial rapid Ag tests for the detection of SARS-CoV-2. Furthermore, the MESIA technique and automated portable device provided results with improved clarity in 15 min as well as reliable semi-quantitative measurement. These results indicate that these rapid Ag tests can be useful for preventing the spread of COVID-19 via timely diagnosis and subsequent containment measures.

## Acknowledgments

We thank the entire laboratory staff for their efforts engaged in the performance of rapid and accurate tests during the COVID-19 pandemic.

## Author Contributions

**Conceptualization:** Sung Jin Jo, Seungok Lee, Jehoon Lee.

**Data curation:** Sung Jin Jo, Sang-hyun Shin, Jungrok Kim, Seungok Lee.

**Formal analysis:** Sung Jin Jo, Seungok Lee.

**Investigation:** Sang-hyun Shin, Jungrok Kim.

**Methodology:** Sung Jin Jo.

**Supervision:** Jehoon Lee.

**Validation:** Seungok Lee, Jehoon Lee.

**Writing – original draft:** Sung Jin Jo.

**Writing – review & editing:** Sung Jin Jo, Sang-hyun Shin, Jungrok Kim, Seungok Lee, Jehoon Lee.

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
