## [Decision Letter · Decision Letter 0]

3 Aug 2021

PONE-D-21-20548

Evaluation of the clinical performance of a magnetic force-assisted electrochemical immunoassay for the detection of SARS-CoV-2 antigens

PLOS ONE

Dear Dr. Lee,

Thank you for submitting your manuscript to PLOS ONE. After careful consideration, we feel that it has merit but does not fully meet PLOS ONE’s publication criteria as it currently stands. Therefore, we invite you to submit a revised version of the manuscript that addresses the points raised during the review process.

Please describe the limitation of your present study, which was commented by one of the reviewers.

We look forward to receiving your revised manuscript.

Kind regards,

Etsuro Ito

Academic Editor

PLOS ONE

Journal Requirements:

2. Please upload a new copy of Figure 1 as the detail is not clear. Please follow the link for more information: " ext-link-type="uri" xlink:type="simple">https://blogs.plos.org/plos/2019/06/looking-good-tips-for-creating-your-plos-figures-graphics/"
https://blogs.plos.org/plos/2019/06/looking-good-tips-for-creating-your-plos-figures-graphics/

3. Please expand the acronym “BBB” (as indicated in your financial disclosure) so that it states the name of your funders in full.

Reviewers' comments:

Reviewer's Responses to Questions

**Comments to the Author**

1. Is the manuscript technically sound, and do the data support the conclusions?

Reviewer #1: Yes

Reviewer #2: Partly

Reviewer #3: Partly

2. Has the statistical analysis been performed appropriately and rigorously? 

Reviewer #1: Yes

Reviewer #2: Yes

Reviewer #3: Yes

3. Have the authors made all data underlying the findings in their manuscript fully available?

Reviewer #1: Yes

Reviewer #2: No

Reviewer #3: Yes

4. Is the manuscript presented in an intelligible fashion and written in standard English?

Reviewer #1: Yes

Reviewer #2: Yes

Reviewer #3: Yes

5. Review Comments to the Author

Reviewer #1: The Authors evaluated the MARK-B COVID-19 Ag test (MARK-B, BBB, Sungnam, Republic of Korea) that is a portable point-of-care device based on the use of a magnetic force-assisted electrochemical immunoassay (MESIA) designed to detect SARS-CoV-2 antigens. Notably, this newly developed immunoassay, MESIA, reportedly suitable for measuring low concentrations of proteins for cancer detection and with the potential to detect any other biomarkers.

The evaluation was done against RT PCR as the gold standard for SARS COV-2 detection and another immunochromatographic manual antigen assay for SARS CoV-2, showing a significant higher correlation with RT PCR and a significant higher sensitivity over the standard antigen assay. Overall, this data confirms that MESIA may provide enhanced sensitivity for point-of-care devices that involve the detection of SARS-CoV-2.

The study is well conducted, each section is well written and developed, statistical measures correctly applied. There is one general limitation in this study:

The MARK-B test has not been CE/FDA-cleared or approved so far for SARS CoV-2, but these validations are pending. Therefore, this is a clear limitation of the study conducted by the Authors, in terms of extension and commercialization of this device. The Authors should discuss this issue and clarify if there are any on-going news on this important issue which is essential for the scientific community. Moreover, the test is not cited by FIND as the date of today July 15 among the 653 antigen immunoassays for SARS CoV-2 enlisted in the web site https://www.finddx.org/covid-19/pipeline/?section=immunoassays#diag_tab

Is this just for Asia? The Authors must clarify it.

Minor issues

Material and Methods:

The semi-quantitative/quantitation system used by MESIA for their electric signal should be specified by the Authors as they use it in the Figures, as well as the nature of the cut-off value and the type of antigen detected by this system.

Reviewer #2: This paper describes the evaluation of a magnetic force-assisted electrochemical immunoassay -based test compared to other commercial rapid Ag tests by application to clinical samples. Authors have successfully compared the results provided by this device with those from other methodologies applying them to a large number of samples. In my opinion, the information offered in the article is very interesting, the electrochemical sensor has demonstrated a good behavior, but the experimental work and the new data in the article are not relevant from the point of view of scientific research. I consider that the paper should not be published in its current state in PLOs One.

Reviewer #3: This is a clear well-written paper describing the performance of a new Ag test for diagnosis of COVID-19 infection. As an independent of assessment of the test, the publication provides important data to users who may be considering use of the test. I recommend publication after revision to address some minor weaknesses in the paper:

1. The main weakness is associated with the comparison to an existing commercial Ag test and the interpretation of the comparison

a. In the abstract and main text their are locations where the authors describe their paper as comparing their test to commercial antigen tests. These statements should be clarified to indicate that they compared to a (as in one) antigen test, and isn't necessarily reflective of how the test would perform to the range of available tests.

b. Some discussion is warranted regarding the criteria for the selection of the comparator test, and the relative performance of this test. Relative to the many Ag tests that are now available commercially, does the published data for the SDQ test place it at the top, middle or bottom of the range for clinical sensitivity and accuracy?

c. It would be helpful to provide some testing with a reference sample (for example, one of the available viral culture reference samples or a sample of recombinant nucleocapsid protein) that could allow direct comparison of the analytical performance of the antigen assays to each other (and potentially other commercial tests) and determination of whether the clinical performance is a direct reflection of the analytical performance).

2. The volume of UTM used to extract the sample should be provided and, if applicable, a discussion of how the volume of UTM may differ from the volume of extraction buffer used in the standard swab protocols for the MESIA and SDQ tests, and how that may affect performance.

3. Ct should be proportional to Log2(RNA concentration) since it represents the number of RNA doubling steps required to achieve a detectable PCR signal. It is therefore probably most appropriate to plot Log2(MESIA current) vs. Ct, which should provide a slope approaching 1. If the log(current) vs. Ct is not linear, that should be discussed.

4. In the methods section, it isn't very clear how the MESIA technique works. I think readers would provide a little more detail or, alternatively, a reference to where they could go for more details.

6. PLOS authors have the option to publish the peer review history of their article (what does this mean?). If published, this will include your full peer review and any attached files.

Reviewer #1: **Yes: **Valeria Ghisetti, MD, Director of the laboratory of Microbiology and Virology, Amedeo di Savoia Hospital, Turin, Italy

Reviewer #2: No

Reviewer #3: No

---

## [Decision Letter · Decision Letter 1]

27 Sep 2021

Evaluation of the clinical performance of a magnetic force-assisted electrochemical immunoassay for the detection of SARS-CoV-2 antigens

PONE-D-21-20548R1

Dear Dr. Lee,

We’re pleased to inform you that your manuscript has been judged scientifically suitable for publication and will be formally accepted for publication once it meets all outstanding technical requirements.

Kind regards,

Etsuro Ito

Academic Editor

PLOS ONE

Reviewers' comments:

Reviewer's Responses to Questions

**Comments to the Author**

1. If the authors have adequately addressed your comments raised in a previous round of review and you feel that this manuscript is now acceptable for publication, you may indicate that here to bypass the “Comments to the Author” section, enter your conflict of interest statement in the “Confidential to Editor” section, and submit your "Accept" recommendation.

Reviewer #3: All comments have been addressed

2. Is the manuscript technically sound, and do the data support the conclusions?

Reviewer #3: Yes

3. Has the statistical analysis been performed appropriately and rigorously? 

Reviewer #3: Yes

4. Have the authors made all data underlying the findings in their manuscript fully available?

Reviewer #3: Yes

5. Is the manuscript presented in an intelligible fashion and written in standard English?

Reviewer #3: Yes

6. Review Comments to the Author

Reviewer #3: Some minor comments/suggestions based on the response to my original review:

1. The description of the assay technology in your response to the reviewers comments is much clearer than in the edits to the manuscript. I suggest just using the text from the response in the manuscript.

2. The line in the log2 antigen vs. Ct count plot should be calculated using the log2 transformed values and should look like a straight line in the plot. Interestingly, the slope of that line will be about -0.3 indicating that an 8-fold change in nucleic acid levels (3 Ct counts) corresponds to roughly a 2-fold change in the antigen assay signal (roughly a cube root dependence). The table of assay signals for the virus titration also shows roughly a cube root dependence.

7. PLOS authors have the option to publish the peer review history of their article (what does this mean?). If published, this will include your full peer review and any attached files.

Reviewer #3: No

---

## [Editor Report · Acceptance letter]

29 Sep 2021

PONE-D-21-20548R1 

Evaluation of the clinical performance of a magnetic force-assisted electrochemical immunoassay for the detection of SARS-CoV-2 antigens 

Dear Dr. Lee:

I'm pleased to inform you that your manuscript has been deemed suitable for publication in PLOS ONE. Congratulations! Your manuscript is now with our production department. 

Kind regards, 

on behalf of

Prof. Etsuro Ito 

Academic Editor

PLOS ONE